# Reward signal in a recurrent circuit drives appetitive long-term memory formation

Toshiharu Ichinose[1,2], Yoshinori Aso[3], Nobuhiro Yamagata[1,2], Ayako Abe[1], Gerald M Rubin[3], Hiromu Tanimoto[1,2]*

[1]Graduate School of Life Sciences, Tohoku University, Sendai, Japan; [2]Max Planck Institute of Neurobiology, Martinsried, Germany; [3]Janelia Research Campus, Howard Hughes Medical Institute, Ashburn, United States

**Abstract** Dopamine signals reward in animal brains. A single presentation of a sugar reward to *Drosophila* activates distinct subsets of dopamine neurons that independently induce short- and long-term olfactory memories (STM and LTM, respectively). In this study, we show that a recurrent reward circuit underlies the formation and consolidation of LTM. This feedback circuit is composed of a single class of reward-signaling dopamine neurons (PAM-α1) projecting to a restricted region of the mushroom body (MB), and a specific MB output cell type, MBON-α1, whose dendrites arborize that same MB compartment. Both MBON-α1 and PAM-α1 neurons are required during the acquisition and consolidation of appetitive LTM. MBON-α1 additionally mediates the retrieval of LTM, which is dependent on the dopamine receptor signaling in the MB α/β neurons. Our results suggest that a reward signal transforms a nascent memory trace into a stable LTM using a feedback circuit at the cost of memory specificity.

## Introduction

Environmentally relevant information, such as poison or food, can trigger formation of long-term memory (LTM) in a single association event (*Garcia et al., 1955*; *Alexander et al., 1984*). Because LTM formation is energetically costly (*Mery and Kawecki, 2005*; *Placais and Preat, 2013*), it is advantageous for animals to refer to surrounding conditions when selecting what information to retain as stable LTM. Appetitive olfactory learning in *Drosophila melanogaster* provides an opportunity to uncover the mechanisms underlying such integration of information, as a single presentation of nutritious sugar and an odor drives robust short-term memory (STM) and LTM, while non-nutritious sugar induces only STM (*Krashes and Waddell, 2008*; *Colomb et al., 2009*; *Burke and Waddell, 2011*).

In *Drosophila*, sugar reward is mediated by a group of dopamine neurons called the protocerebral anterior medial (PAM) cluster (*Burke et al., 2012*; *Liu et al., 2012*). Artificial activation of the PAM cluster neurons in the presence of odor drives robust appetitive STM and LTM formation. Recent studies subdivided the PAM cluster neurons into STM- and LTM-inducing cell types and identified several cell types responsible for LTM formation, including PAM-α1 (*Huetteroth et al., 2015*; *Yamagata et al., 2015*). Interestingly, the induced LTM is undetectable immediately after conditioning but gradually develops with time (*Das et al., 2014*; *Huetteroth et al., 2015*; *Yamagata et al., 2015*).

Dopamine reward signals are conveyed by PAM cluster neurons to a brain structure called mushroom body (MB), where the convergence of the reward and the odor is assumed to take place (*Schwaerzel et al., 2003*; *Krashes et al., 2007*; *Trannoy et al., 2011*; *Cervantes-Sandoval et al., 2013*; *Huang et al., 2013*). The MB represents odors using a sparse subset of ~2000 intrinsic neurons, Kenyon cells (KCs) (*Turner et al., 2008*; *Aso et al., 2009*; *Honegger et al., 2011*; *Campbell et al., 2013*). KCs send parallel fibers to form lobe structures, where multiple MB-output neuron (MBON) types elaborate spatially segregated dendritic arbors (*Aso et al., 2014a*). The KC-MBON synapses are

*For correspondence:
hiromut@m.tohoku.ac.jp

Competing interests: The authors declare that no competing interests exist.

**eLife digest** An animal that finds particularly nutritious and palatable food will often develop a long-lasting memory—even if they experience that event only once.

One example of this is the ability of the fruit fly *Drosophila* to form a long-term association between a sugar reward and a specific odor that was present when they received the reward. The consumption of sugar triggers the release of a chemical called dopamine on specific compartments of a brain structure called the mushroom body. Dopamine then acts to modify the connection between cells called "Kenyon cells", which encode specific odors, and the neurons that send signals out from the mushroom body (called MBONs). The result is the formation of a memory that links the odor with the reward.

However, little is known about how this process differs for long-term vs. short-term memories, and how it can occur when the fly has experienced the odor and reward together on only a single occasion. To find out, Ichinose et al. combined behavioral testing of fruit flies with genetics. The results confirmed that the dopamine neurons and the MBONs that project to a single compartment of the mushroom body, called α1, are both required for the formation of long-term odor-reward memories, but not their short-term equivalents. These neurons are called PAM-α1 and MBON-α1, respectively.

Unexpectedly, anatomical data revealed that PAM-α1 dopamine neurons receive input from MBON-α1; that is, long-term memory formation involves a feedback circuit: from PAM-α1 to Kenyon cells, then to MBON-α1 and back to PAM-α1. Blocking feedback from the MBON-α1 onto the PAM-α1 neurons shortly after odor-reward training disrupted long-term memory formation. Conversely, blocking feedback at a later stage did not. This suggests that prolonged activation of PAM-α1 by MBON-α1 helps to strengthen newly established memories, converting them into memories that will last for a long time.

The discovery of a specific circuit that supports long-term, but not short-term, memory formation in fruit flies is consistent with evidence of distinct mechanisms underlying these processes in mammals. Further work is now required to determine whether feedback circuits similar to those in fruit flies also contribute to reward-based learning in other animals.

presumed to be modified in the course of memory formation by the reinforcing dopamine neurons (*Tomchik and Davis, 2009*; *Gervasi et al., 2010*; *Sejourne et al., 2011*; *Cervantes-Sandoval et al., 2013*; *Placais et al., 2013*; *Boto et al., 2014*).

While there has been progress on the molecular and cellular mechanisms underlying LTM formation resulting from repetitive conditioning (*Pagani et al., 2009*; *Placais et al., 2012*; *Philips et al., 2013*), how a single presentation of a reward can trigger LTM remains to be understood. Here, we identify and behaviorally characterize a recurrent reward circuit that is essential for appetitive LTM formation and consolidation. The recurrent circuit consists of PAM-α1 dopamine neurons and a single MBON type, MBON-α1. This recurrent dopamine reward circuit may provide insights into a general mechanism for LTM formation.

## Results

### MBON-α1 provides a glutamatergic feedback to PAM-α1

PAM-α1 delivers reward signals essential for appetitive LTM formation (*Yamagata et al., 2015*). To understand the neuronal circuit that regulates rewards for LTM formation, we first sought to anatomically identify the input neurons to PAM-α1. The presynaptic terminals of PAM-α1 are localized to the basal compartment of the MB α lobe (α1) (*Liu et al., 2012*), whereas the dendrites stretch horizontally across the superior medial, intermediate, and lateral protocerebra (SMP, SIP, and SLP, respectively) (*Aso et al., 2014a*; *Yamagata et al., 2015*) (*Figure 1A,B,H*). By segmenting the dendrites of PAM-α1 in confocal images (*Aso et al., 2014a*), we queried a database of GAL4 expression patterns (*Jenett et al., 2012*) to computationally search for candidate upstream neurons. Strikingly, one of the input candidates we identified was the output neurons from the α1 compartment of the MB (MBON-α1) (*Figure 1C,D*). Transgenic expression of a presynaptic marker (*Robinson et al., 2002*) by *MB310C-GAL4*

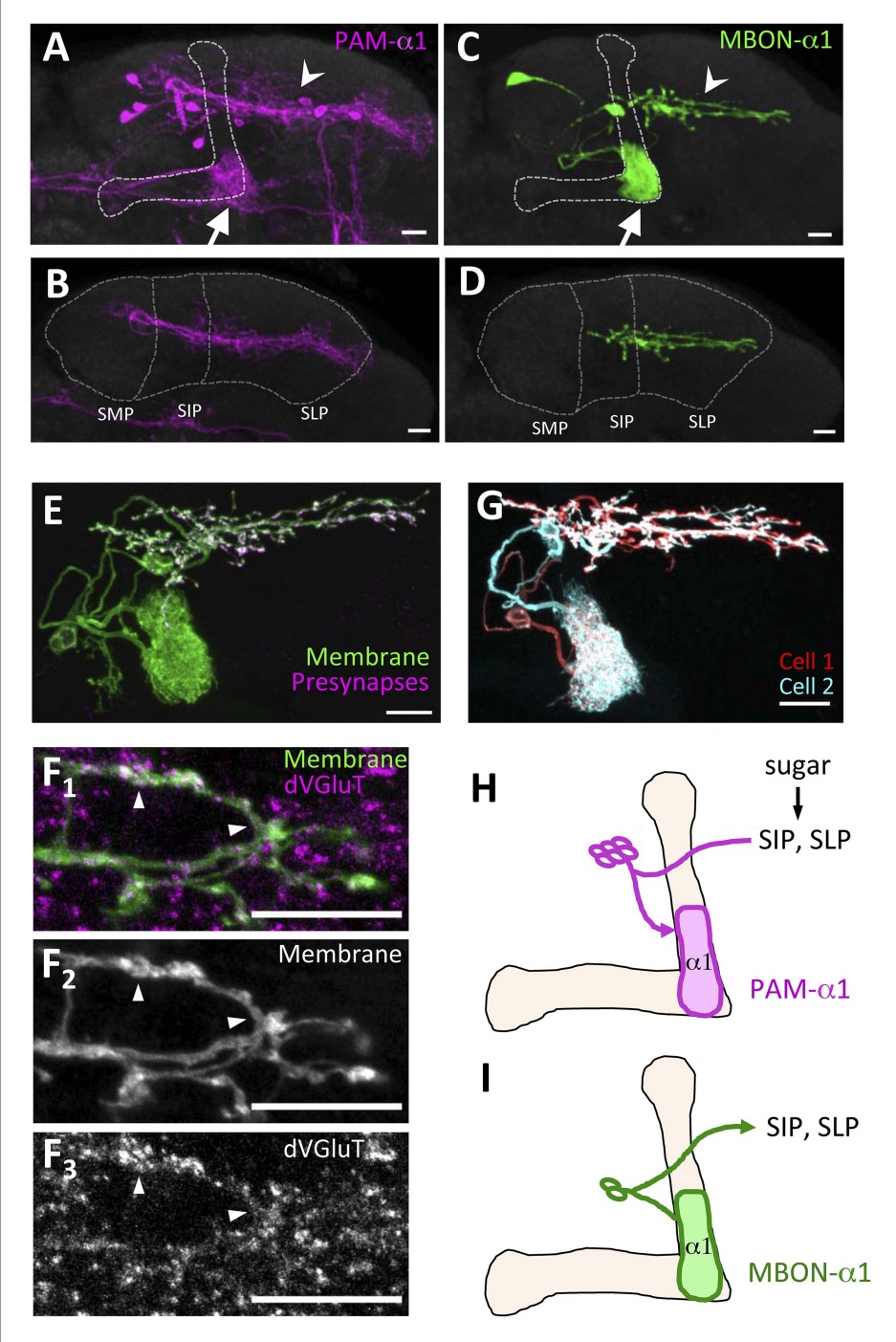

**Figure 1**. MBON-α1 receives inputs from MB-α1 and projects to the dendrites of PAM-α1. (**A–D**) Anatomy of PAM-α1 (**A**, **B**) and MBON-α1 (**C**, **D**). Dotted line indicates α/β lobes in the mushroom body (MB) (**A**, **C**) or SMP, SIP, and SLP (superior medial, intermediate, and lateral protocerebra, respectively) (**B**, **D**). Arrows indicate α1 and arrowheads the SIP and SLP (**A**, **C**). PAM-α1 and MBON-α1 are visualized by *pJFRC2-10xUAS-mCD8GFP* in *VK00005* and *MB299B-GAL4* (**A**, **B**) or *MB310C-GAL4* (**C**, **D**), respectively. (**E**) Presynaptic terminals of MBON-α1 are highly localized in SIP and SLP. *MB310C-GAL4* is used to drive a general membrane marker (green) and a presynaptic marker (magenta) in MBON-α1. (**F**) Double labeling of the membrane of MBON-α1 (green) and anti-vesicular glutamate transporter (dVGluT, magenta) (**F₁**), membrane staining (**F₂**), and anti-dVGluT staining (**F₃**). Arrowheads highlight the overlap. (**G**) Two individual MBON-α1 neurons visualized by multi-color flip-out with different colors (red and cyan). (**H**, **I**) Schematics of PAM-α1 and MBON-α1, respectively. The α/β lobe of the MB is outlined with light orange. Scale bars, 10 μm.

*Figure 1. continued on next page*

*Figure 1. Continued*

The following figure supplement is available for figure 1:

**Figure supplement 1**. MBON-α1 is neither GABAergic nor cholinergic.

revealed that the presynaptic terminals of MBON-α1 lie in close apposition to the dendrites of PAM-α1 (*Figure 1E,I*). The terminals of MBON-α1 were immunoreactive for the vesicular glutamate transporter (*Figure 1F*) (*Daniels et al., 2008*) but not for markers for GABAergic or cholinergic neurons (glutamic acid decarboxylase [*Jackson et al., 1990*] or choline acetyltransferase [*Takagawa and Salvaterra, 1996*], respectively) (*Figure 1—figure supplement 1*), consistent with MBON-α1 being glutamatergic (*Aso et al., 2014a*). Differential labeling of the individual MBON-α1 neurons in the same brain (*Nern et al., 2015*) revealed that MBON-α1 comprises two cells with very similar morphology (*Figure 1G*).

To precisely determine the relative arrangement of PAM-α1 and MBON-α1 neurites, we performed a double-labeling experiment using *R72D01-LexA*, which labels PAM-α1, and the split-GAL4 driver *MB310C-GAL4*, which labels MBON-α1 (*Aso et al., 2014a*). Confocal images revealed that the processes of MBON-α1 and PAM-α1 substantially intermingled both in the SIP and SLP, and in the α1 compartment of the MB—the input and output sites of PAM-α1, respectively (*Figure 2A–E*, *Figure 2—figure supplement 1*, *Video 1*). To observe these processes in SIP and SLP in even greater detail, we employed a microscope with a super resolution detection system and found bouton-like structures of MBON-α1 attached to the dendrites of PAM-α1 (*Figure 2B–D*). Furthermore, a GFP reconstitution across synaptic partners (GRASP) (*Feinberg et al., 2008*; *Gordon and Scott, 2009*) experiment using *R72D01-LexA* and *MB310C-GAL4* revealed strong GRASP signals in SIP, SLP, and the α1 compartment (*Figure 2F*). This further supports the close juxtaposition between PAM-α1 and MBON-α1.

## Appetitive LTM formation requires feedback from MBON-α1

If the MB feedback modulates PAM-α1, the output of MBON-α1 might be required for the acquisition of LTM of the sugar reward. We examined the requirement of PAM-α1 and MBON-α1 by expressing a temperature-sensitive dominant negative dynamin, Shibire[ts1] (*Kitamoto, 2001*) using the split-GAL4 drivers *MB299B-GAL4* and *MB310C-GAL4* (*Figure 3B,F*). *MB299B-GAL4* has strong expression in PAM-α1 (*Aso et al., 2014a*; *Yamagata et al., 2015*) (*Figure 3B*). Consistent with our previous results (*Yamagata et al., 2015*), the blockade of PAM-α1 during conditioning did not significantly affect STM (*Figure 3C*), but led to a severe impairment of sucrose-rewarded LTM (*Figure 3D*). The unimpaired performance in appetitive STM demonstrates that flies of these genotypes have normal sugar and odor perception at the restrictive temperature. Intriguingly, we found a similar preferential impairment of LTM upon the blockade of MBON-α1, using *MB310C-GAL4* driver (*Figure 3H*). The same blockade did not significantly impair STM (*Figure 3G*). We confirmed the preferential requirement of MBON-α1 for LTM using another driver, *MB323B-GAL4* (*Figure 3—figure supplement 1A*), and obtained similar results (*Figure 3G,H*); neither *MB310C/UAS-shi* nor *MB323B/UAS-shi* flies showed a significant LTM impairment when trained and tested at the permissive temperature (*Figure 3—figure supplement 1B*).

We next examined whether glutamatergic input to PAM-α1 is required for LTM formation by down-regulating the expression of glutamate receptors using *MB299B-GAL4*. For efficient and cell-type-specific manipulation, we used *UAS-RNAi* fly strains based on the shRNA technique (*Ni et al., 2011*). Intriguingly, knocking down either *dNR-1* or *dNR-2*, which together form the functional NMDA receptor (*Xia et al., 2005*), impaired LTM (*Figure 4C*) but not STM (*Figure 4B*). This preferential requirement in LTM is similar to the blockade of MBON-α1 and PAM-α1, suggesting the direct glutamatergic feedback.

Given that thermogenetic stimulation of PAM-α1 with a temperature-sensitive cation channel *dTrpA1* (*Hamada et al., 2008*) can provide the reward signal for LTM formation (*Yamagata et al., 2015*), we asked if activation of MBON-α1 would also be capable of inducing appetitive LTM (*Figure 5A*). Indeed, we found appetitive LTM upon thermo-activation of MBON-α1 in the presence of an odor (*Figure 5B*). These results, together with the consistent requirements of MBON-α1 and NMDAR-signaling in PAM-α1, are most easily explained if the glutamatergic feedback regulation from MBON-α1 augments dopamine reward signaling by PAM-α1.

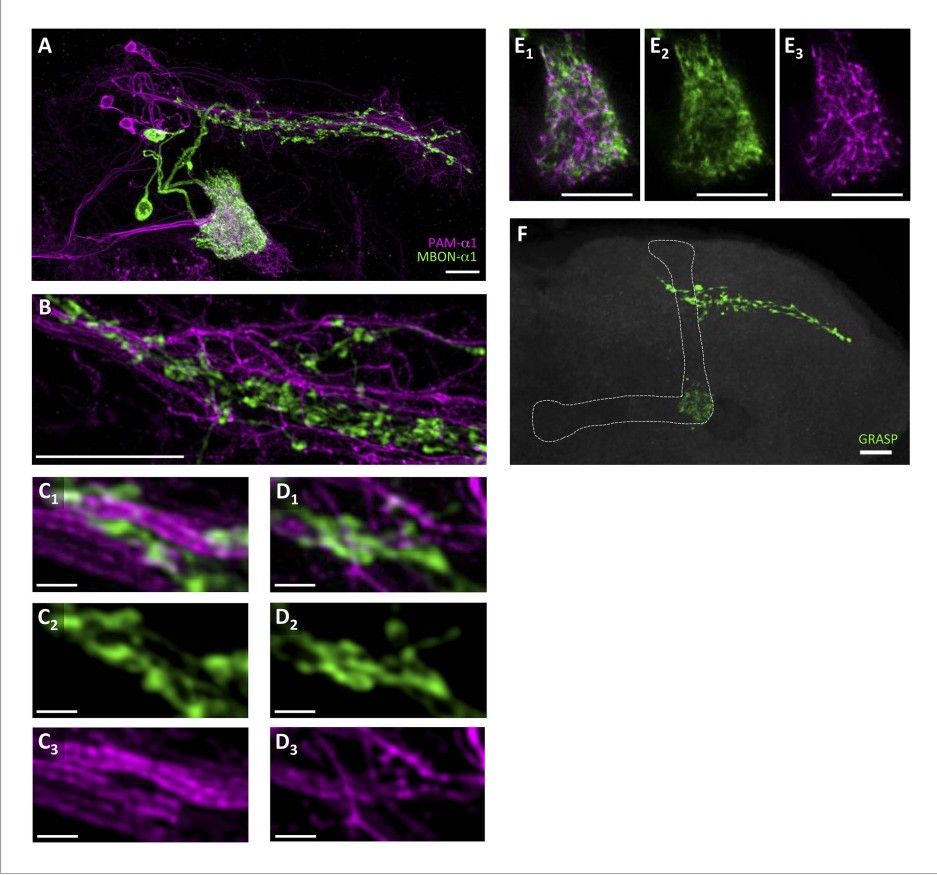

**Figure 2**. Processes of PAM-α1 and MBON-α1 intermingle each other. (**A**) Double labeling of PAM-α1 and MBON-α1. PAM-α1 (magenta) and MBON-α1 (green) are visualized using *R72D01-LexA* and *MB310C-GAL4*, respectively. (**B**–**D**) Magnified substack images in SIP and SLP, obtained by a super-resolution detection system. (**E**) Magnified substack image in the α1 compartment in the MB. (**F**) GFP reconstitution across synaptic partner (GRASP) signals in SIP, SLP, and MB α1 support the contacts between PAM-α1 and MBON-α1. Scale bars, 10 μm (**A**, **B**, **E**, **F**), 1 μm (**C**, **D**).

The following figure supplement is available for figure 2:

**Figure supplement 1**. Stereotyped projections of PAM-α1 and MBON-α1.

## *DopR1*-mediated appetitive LTM trace is formed in α/β KCs

Several lines of evidence suggest that D1-like dopamine receptor (*DopR1*) is required in KCs for appetitive memory (*Kim et al., 2007*; *Liu et al., 2012*; *Boto et al., 2014*). However, previous studies focused on only STM. To identify the receiving site of dopamine for LTM formation, we knocked down *DopR1*, either in α/β KCs or in MBON-α1, the potential targets of dopamine from PAM-α1, and measured appetitive LTM. To target the expression specifically to the α/β KCs, we employed two driver lines, *MB008B-GAL4* (*Aso et al., 2014a*; *Vogt et al., 2014*) (*Figure 6—figure supplement 1A*) and *c739-GAL4* (*Yang et al., 1995*). Interestingly, knocking down *DopR1* in the α/β KCs, using *c739-GAL4* or *MB008B-GAL4*, resulted in a severe impairment of appetitive LTM (*Figure 6A*) while leaving STM unaffected (*Figure 6—figure supplement 1B*). These results are consistent with the observation that the coincidence detector adenylate cyclase encoded by *rutabaga*, thought to act downstream of the dopamine receptor (*Tomchik and Davis, 2009*; *Gervasi et al., 2010*; *Boto et al., 2014*), functions for LTM in the α/β KCs (*Trannoy et al., 2011*). On the other hand, knocking down *DopR1* in MBON-α1 did not significantly impair either appetitive LTM or STM (*Figure 6A* and *Figure 6—figure supplement 1B*).

To further localize the role of *DopR1* in KCs, we performed a rescue experiment, using a piggyBac insertion mutant, *dumb²* (*Liu et al., 2012*; *Qin et al., 2012*). *dumb²* mutant flies showed severely impaired appetitive LTM. Targeting wild-type *DopR1* gene expression in KCs fully rescued the defects

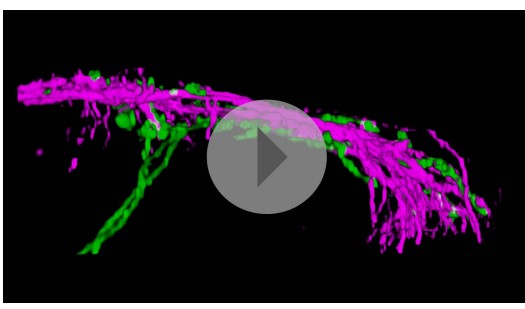

**Video 1.** Volume-rendered image of processes of PAM-α1 (magenta) and MBON-α1 (green) in SIP and SLP. DOI: 10.7554/eLife.10719.007

(*Figure 6B*). Together, although we cannot exclude the possible role of other dopamine receptors in MBON-α1, we propose that *DopR1-*mediated memory trace is formed in α/β KCs not in MBON-α1.

## Output from the α/β KCs is required during conditioning

Previous studies have highlighted the role of the output from α/β KCs for memory retrieval (*Krashes et al., 2007*; *Trannoy et al., 2011*; *Cervantes-Sandoval et al., 2013*; *Perisse et al., 2013*). Given the requirement of MBON-α1 for the acquisition of appetitive LTM, the output of the α/β KCs during conditioning may be important for LTM formation. To test this possibility, we blocked the output of the α/β KCs with Shi[ts1]. Strikingly, the blockade of the α/β KCs during conditioning significantly impaired appetitive LTM, but not STM (*Figure 7A,B*), similar to the blockade of PAM-α1 and MBON-α1 (*Figure 3*). When the flies were trained and tested at the permissive temperature, no significant impairment of appetitive LTM was observed (*Figure 7—figure supplement 1*). The requirement for α/β KC synaptic signaling during LTM formation supports our model that MBON-α1 provides information about the memory trace, formed in the α/β KCs, to the reward signaling PAM-α1 neurons.

## LTM retrieval requires MBON-α1

If the nascent memory trace in α/β KCs is communicated by MBON-α1 during conditioning, MBON-α1 might also convey the read-out of LTM. Intriguingly, when MBON-α1 was blocked only during the test phase of LTM, we found a severe impairment of conditioned odor approach (*Figure 8A*). In contrast, the blockade of PAM-α1 during LTM retrieval did not have a significant effect (*Figure 8B*). These results imply that MBON-α1 plays a dual role in both formation and retrieval of LTM; different post-synaptic targets of MBON-α1 would be used in these two roles, with the recurrent PAM-α1-MBON-α1 circuit selectively involved in LTM formation.

## Post-training dopamine release consolidates the nascent memory trace at the cost of memory specificity

The glutamatergic input from MBON-α1 might provide an excitatory feedback on the PAM-α1 neurons via the NMDA receptor thereby prolonging the dopamine release after the cessation of sugar presentation. If this model were correct, the post-training blockade of the feedback circuit might be expected to impair appetitive LTM. Consistent with this idea, blocking any component of the feedback circuit for 1 hr immediately following conditioning resulted in a significant impairment of appetitive LTM (*Figure 9A,C,E*), whereas the same blockade 22 hr after training did not (*Figure 9B,D, F*). These results strongly suggest that the recurrent circuit has an ongoing activity, which plays a key role in early phase of appetitive LTM consolidation.

Prolonged dopamine release in the absence of odor may also modulate KCs that do not respond to the rewarded odor. If so, the odor specificity of appetitive LTM might be compromised compared to STM. Thus, we measured the generalization profile of STM and LTM by testing with the graded mixtures of 'contaminating' odors (*Figure 10*). Odor mixtures were created by systematically varying the relative contents of the contaminant (trained odor: contaminant = 100:0, 80:20, 60:40, 40:60, 20:80, or 0:100): if the memory is specific to the trained odor, the performance quickly decreases with the increasing contaminant ratio. Strikingly, the performance decline is shallower in LTM compared to STM (*Figure 10C,D*). We confirmed these results with another set of odorants, suggesting that compromised stimulus specificity in LTM is general (*Figure 10—figure supplement 1*). Considering that the only difference between these groups is the retention time, there may be a trade-off between consolidation and specificity, possibly due to the prolonged dopamine release in the MB following memory acquisition.

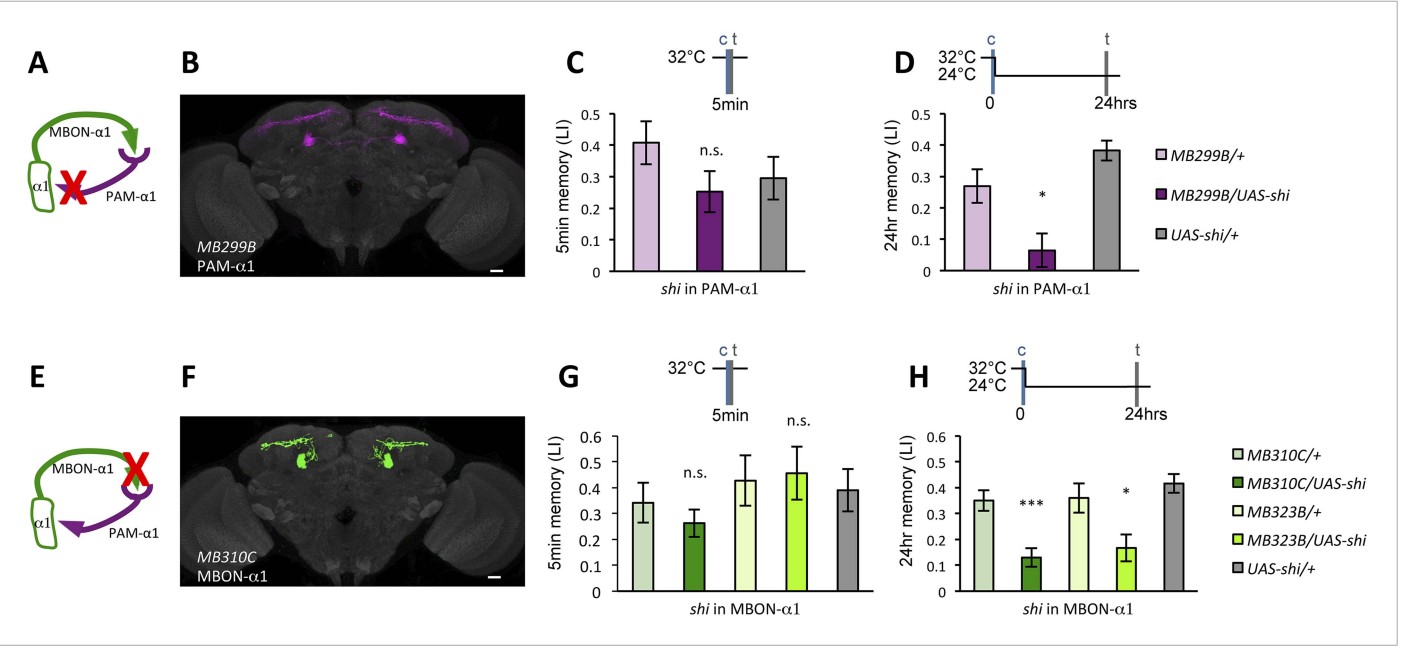

**Figure 3**. Feedback from MBON-α1 is required for appetitive LTM formation. (**A**) Diagram of the experiment. (**B**) Expression pattern of *MB299B-GAL4*. (**C**) Blockade of PAM-α1 does not impair STM significantly. 5-min appetitive memory was measured (*n* = 10, 10, 12). c: conditioning, t: test. (**D**) Blockade of PAM-α1 during conditioning impairs LTM. 24-hr appetitive LTM was measured (*n* = 9, 12, 12). (**E**) Diagram of experiment. (**F**) Expression pattern of *MB310C-GAL4*. (**G**) Blockade of MBON-α1 does not impair STM significantly (*n* = 6, 7, 10, 6, 10). (**H**) Blockade of MBON-α1 during conditioning impairs LTM. *MB323B-GAL4* is a second driver line that expresses in MBON-α1 (see figure supplement; *n* = 23, 24, 13, 14, 24). Bar graphs are mean ± s.e.m. *: p < 0.05, ***: p < 0.001, n.s.: p > 0.05. Scale bars, 20 μm.

The following figure supplement is available for figure 3:

**Figure supplement 1**. (**A**) Expression pattern of MB323B-GAL4.

# Discussion

## Function of a recurrent reward circuit

Prior studies on the role of MBONs in learning and memory have focused on memory retrieval (*Sejourne et al., 2011*; *Pai et al., 2013*; *Placais et al., 2013*; *Aso et al., 2014b*). Here, we extend the known role of MBONs and show that MB output through MBON-α1 is required not only for the retrieval, but also for the acquisition and early consolidation of long-term appetitive olfactory memory. This unanticipated aspect of the MB output for LTM formation appears to function through a recurrent loop (*Figure 11*): MBON-α1 dendrites and the terminals of PAM-α1 occupy the same MB compartment (*Aso et al., 2014a*); the *DopR1* dopamine receptor is required in the KCs of that compartment (*Figure 6*); MBON-α1 in turn sends a portion of its outputs to the dendrites of PAM-α1 (*Figures 1, 2*) closing the loop; synaptic signaling by all three cell types—α/β KCs, MBON-α1, and PAM-α1—is required during LTM formation and consolidation (*Figures 3, 7, 9*). These findings strongly support our model that feedback signals from α/β KCs, mediated by MBON-α1, play a critical role in controlling the rewarding dopamine signals from PAM-α1 (*Figure 11*). Inhibitory recurrent circuits in the MBs have been implicated in appetitive learning of honeybees (*Grunewald, 1999a*, *1999b*). Our present study reveals a new circuit motif, a feedback circuit that is required for dopamine neurons to provide reinforcement during appetitive LTM formation.

Dopamine has been thought to simply relay rewarding or punitive stimuli (*Gerber et al., 2004*; *Waddell, 2013*). However, our results call into question this prevailing circuit model. More specifically, PAM-α1 might integrate simultaneous inputs from sugar perception and the nascent memory trace to augment and sustain the dopamine release to gate LTM formation (*Figure 11*). The requirement of

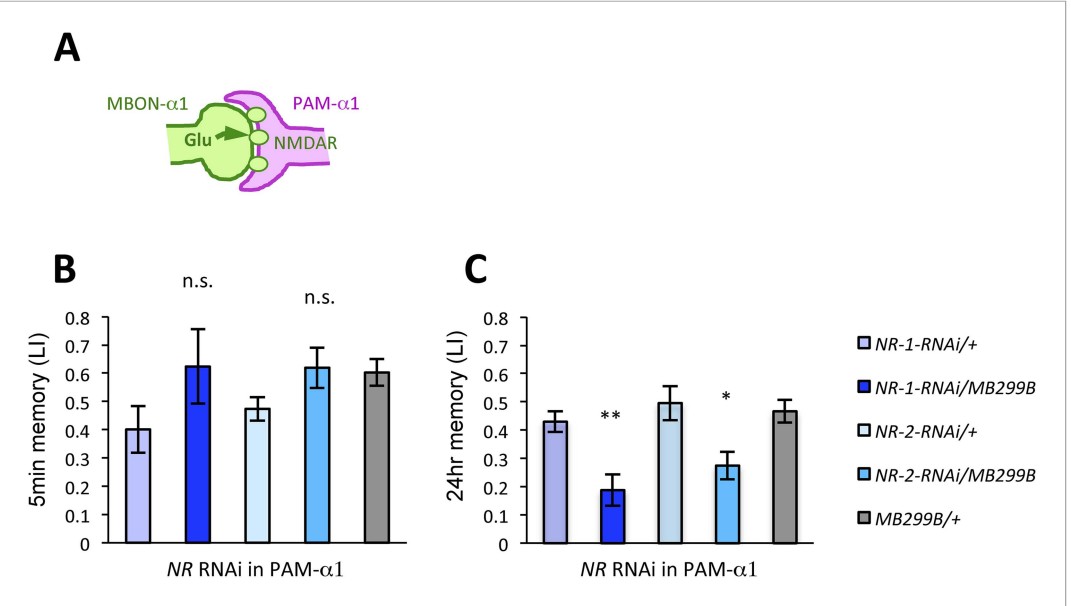

Figure 4. NMDA receptors in PAM-α1 are required for appetitive LTM. (A) The subunits of NMDA receptor are down-regulated in PAM-α1. (B, C) Knocking down NMDA receptor subunits does not impair 5-min memory (B: $n$ = 16, 8, 13, 12, 13) but impairs 24-hr memory significantly (C: $n$ = 18, 8, 22, 20, 29). Bar graphs are mean ± s.e.m. *: $p < 0.05$, **: $p < 0.01$, n.s.: $p > 0.05$. Scale bars, 20 μm.

the NMDA receptor in PAM-α1 (*Figure 4*) further supports this model. Such integration by PAM-α1 might enhance the reward gain selectively in relevant sensory environments. In other words, the reinforcing signal for LTM formation may depend on the presence of other sensory stimuli (e.g., odor). As LTM formation is energetically costly (*Mery and Kawecki, 2005*; *Placais and Preat, 2013*; *Musso et al., 2015*), this gain control of the reinforcing signal might serve to restrict LTM formation to ecologically relevant situations. Although the requirement of NMDA receptor in PAM-α1 implies that MBON-α1 provides excitatory input, the synaptic connectivity between MBON-α1 and PAM-α1 awaits physiological demonstration, given functional inhibitory glutamate receptor signaling (*Liu and Wilson, 2013*).

In addition to the role in memory acquisition, the PAM-α1 output is required during consolidation (*Figure 9A,B*). The dopamine release is likely to be sustained by the ongoing activity of the recurrent circuit (*Figure 9C–F*). A recent study reported a consistent finding that the output from α/β KCs during consolidation has an essential role for appetitive and aversive LTM (*Huang et al., 2012*) in addition to their role in retrieval (*Krashes et al., 2007*; *Trannoy et al., 2011*; *Cervantes-Sandoval et al., 2013*; *Xie et al., 2013*). Recent studies revealed a gradual development of appetitive LTM after conditioning (*Das et al., 2014*; *Huetteroth et al., 2015*; *Yamagata et al., 2015*). Ongoing activity in the

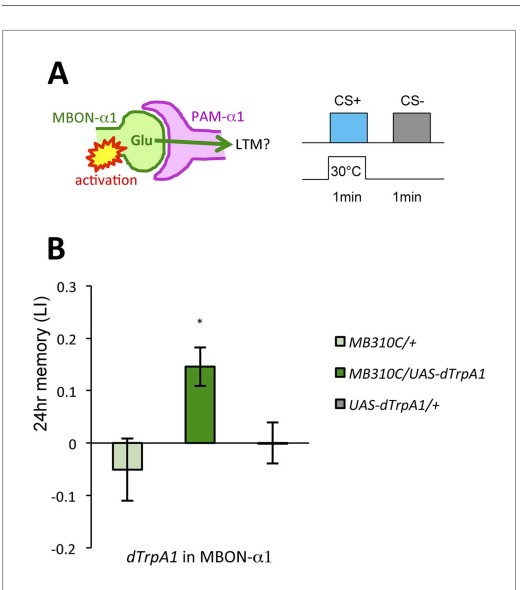

Figure 5. MBON-α1 signals reward for appetitive LTM. (A) Activation of MBON-α1 was paired with odor presentations and 24-hr memory was measured. (B) Activation of MBON-α1 induces appetitive LTM formation ($n$ = 16, 24, 24). Bar graphs are mean ± s.e.m. *: $p < 0.05$, n.s.: $p > 0.05$. Scale bars, 20 μm.

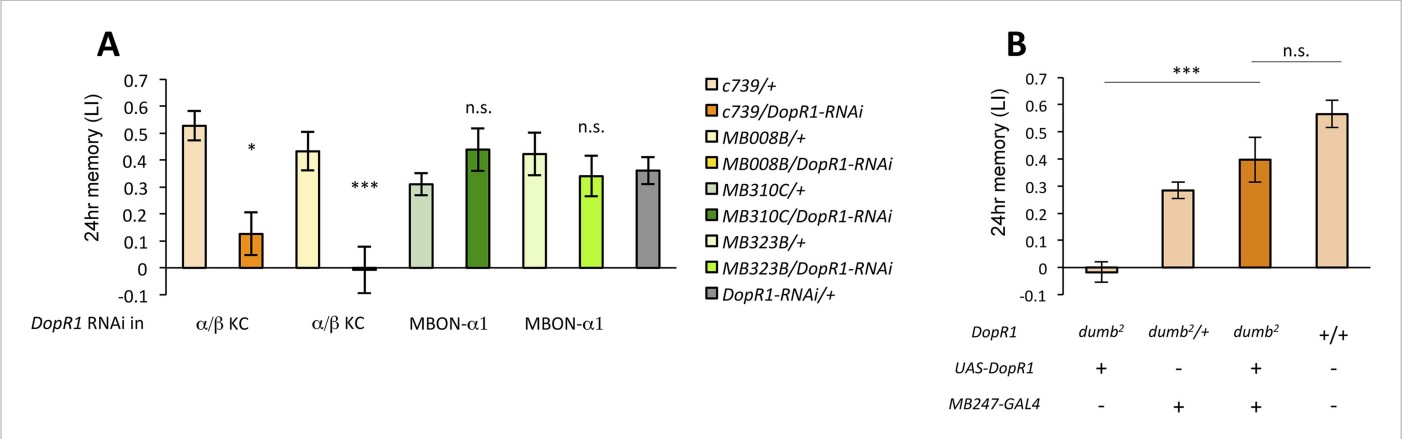

Figure 6. α/β KCs receive dopamine signals through DopR1 for LTM formation. (A) Knocking down *DopR1* in α/β Kenyon cells (KCs) but not in MBON-α1 impairs appetitive LTM (*n* = 17, 10, 11, 11, 18, 12, 7, 12, 27). (B) KC expression of *DopR1* in the *dumb²* mutant background fully rescues the LTM impairment (*n* = 12, 10, 7, 13). Bar graphs are mean ± s.e.m. *: p < 0.05, ***: p < 0.001.

The following figure supplement is available for figure 6:

Figure supplement 1. (A) Expression pattern of MB008B-GAL4.

recurrent loop we propose here may provide a circuit mechanism for the gradual LTM development. In addition, previous studies reported that LTM consolidation in *Drosophila* (*Placais et al., 2012*; *Musso et al., 2015*) or in rats (*Rossato et al., 2009*) requires post-training dopamine inputs.

Our generalization experiments show that appetitive LTM is less specific to the trained odor, compared to STM (*Figure 10*). This result fits well into previous studies in mice and bees, demonstrating that long-lasting memories are more generalized (*Stach and Giurfa, 2005*; *Wiltgen and Silva, 2007*). More generalizable LTM makes sense from an ecological viewpoint: it is worth taking a chance for survival even if the smell is not exactly the same as learned.

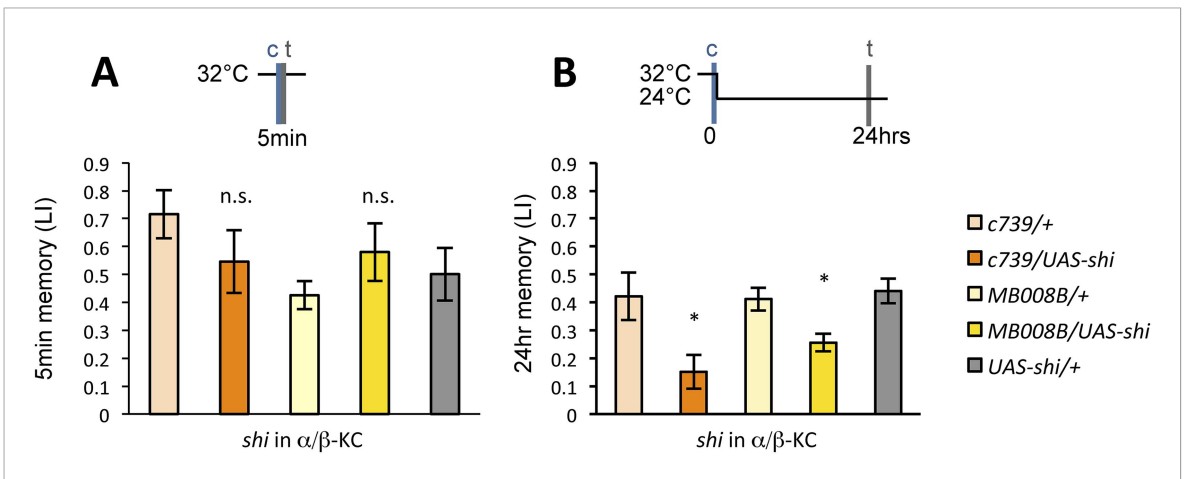

Figure 7. Output of α/β KCs has an essential role in the acquisition of appetitive LTM. (A) Blockade of α/β KCs does not impair STM significantly (*n* = 8, 7, 7, 8, 9). (B) Blockade of α/β KCs during conditioning impairs LTM (*n* = 7, 7, 18, 19, 20). Bar graphs are mean ± s.e.m. *: p < 0.05, n.s.: p > 0.05.
The following figure supplement is available for figure 7:

Figure supplement 1. Training and testing *c739/UAS-shi* and *MB008B/UAS-shi* flies at the permissive temperature do not impair appetitive LTM (*n* = 7, 6, 8, 9, 6).

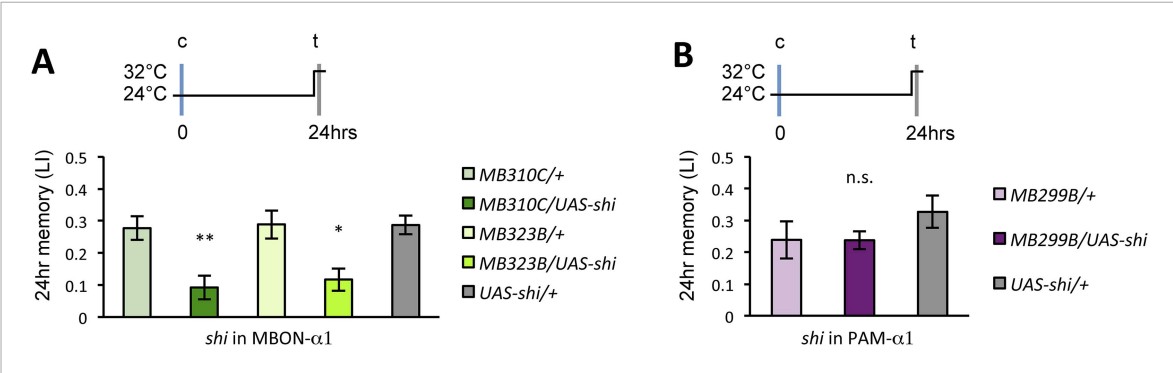

**Figure 8.** Appetitive LTM is read out through MBON-α1. (**A**) Blocking MBON-α1 during test impairs appetitive LTM retrieval (*n* = 20, 21, 11, 13, 20). (**B**) Blocking PAM-α1 during test does not significantly impair LTM retrieval (*n* = 7, 7, 10). Bar graphs are mean ± s.e.m. *: p < 0.05, **: p < 0.01, n.s.: p > 0.05.

In the fly brain, distinct types of dopamine neurons signal reward for STM and LTM, and these memories are independently formed in different MB lobes (*Trannoy et al., 2011*; *Huetteroth et al., 2015*; *Yamagata et al., 2015*). Octopamine has been widely accepted as a positive reinforcement signal in insects (*Hammer, 1993*; *Hammer and Menzel, 1998*; *Schwaerzel et al., 2003*; *Schroll et al., 2006*; *Perry and Barron, 2013*). In *Drosophila*, it is considered to provide inputs to dopamine neurons that induce STM formation (*Burke et al., 2012*; *Das et al., 2014*; *Huetteroth et al., 2015*). Recent studies in mammals reached a similar conclusion that memories with different temporal dynamics are mutually conflicting and formed in distinct areas, to which different classes of dopamine neurons project (*Liljenstrom, 2003*; *Abraham and Robins, 2005*; *Kim and Hikosaka, 2013*; *Kim et al., 2014*). Independent processing of STM and LTM might be a conserved strategy among animals.

## Circuit mechanism of memory-guided behavior

In addition to the function in the recurrent reward circuit, MBON-α1 mediates retrieval of the LTM trace in the MB (*Figure 8*). MBON-α1 samples from the MB-α1 and terminates in the SIP and the SLP. Another type of MBONs also mediates appetitive LTM retrieval: MBON-α3 (also known as MB-V3) (*Placais et al., 2013*). In contrast to MBON-α1, blocking MBON-α3 during conditioning does not impair appetitive LTM acquisition (*Placais et al., 2013*). MBON-α3 arborizes dendrites in the tip of the α lobe and projects to the SMP and the SIP where the terminals of MBON-α1 and MBON-α3 are intermingled (*Aso et al., 2014a*). Therefore, appetitive LTM formed in the α/β KCs is read out through at least two different classes of MBONs. These MBONs converge in the SIP (*Aso et al., 2014a*) and might activate shared downstream targets to guide conditioned odor approach.

## Interplay between reinforcing signals and learned information

The recent comprehensive connectivity map of the afferent and efferent neurons of the *Drosophila* MB revealed many potential feedback connections outside the MB (*Aso et al., 2014a*). Considering the central role of the MB in sensory integration and memory-guided behavior (*Aso et al., 2014b*), recurrent computation between learned information and reinforcement signals might be a fundamental principle of memory processing.

In both vertebrates and invertebrates, the activity of reinforcement neurons is modified upon associative learning (*Hammer, 1993*; *Mirenowicz and Schultz, 1994*; *Riemensperger et al., 2005*; *Cohen et al., 2012*; *Schultz, 2013*). Our results delineated such a feedback circuit, which was predicted by a physiological study in *Drosophila* (*Riemensperger et al., 2005*). In this circuit, MBON-α1 mediates monosynaptic feedback to the dopamine neurons. In contrast, there is no monosynaptic feedback to the dopamine neurons found in mammals, rather anatomical studies suggest such a feedback to be polysynaptic (*Lisman and Grace, 2005*; *Watabe-Uchida et al., 2012*). Further comparative studies should assess how the recurrent circuit motif of reinforcement is implemented in different animal brains.

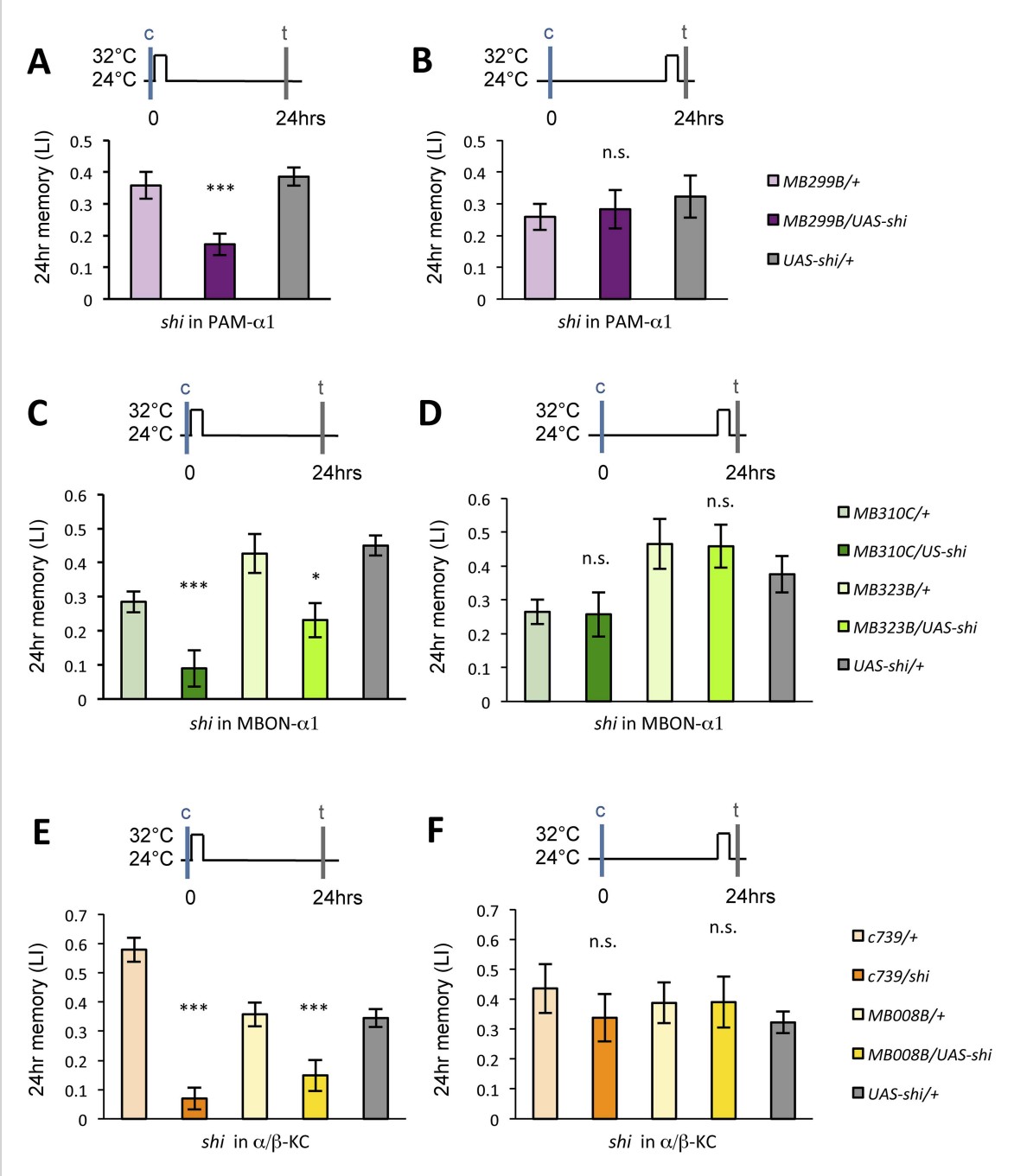

**Figure 9**. Dopamine release from PAM-α1 is required for appetitive LTM consolidation. (**A**, **C**, **E**) Blocking PAM-α1 (**A**), MBON-α1 (**C**), or α/β–KC (**E**) for 1 hr immediately after conditioning impairs appetitive LTM ($n$ = 16, 15, 22 (**A**); $n$ = 15, 16, 14, 20, 34 (**C**); $n$ = 9, 8, 10, 7, 15 (**E**)). (**B**, **D**, **F**) Blocking PAM-α1 (**B**), MBON-α1 (**D**), or α/β–KC (**F**) for 1 hr, 22 hr after conditioning does not impair appetitive LTM ($n$ = 15, 15, 14 (**B**); $n$ = 16, 15, 13, 13, 23 (**D**); n = 12, 12, 15, 12, 15 (**F**)). Bar graphs are mean ± s.e.m. *: p < 0.05, ***: p < 0.001, n.s.: p > 0.05.

## Materials and methods

### Flies

*Canton-S* was used as a wild-type strain. Generation and basic characterization of the split-GAL4 drivers (*MB008B-GAL4*, *MB299B-GAL4*, *MB310C-GAL4*, and *MB323B-GAL4*) are described in *Aso et al. (2014a)*. *R72D01-LexA* was constructed using the methods described in *Pfeiffer et al. (2010)*

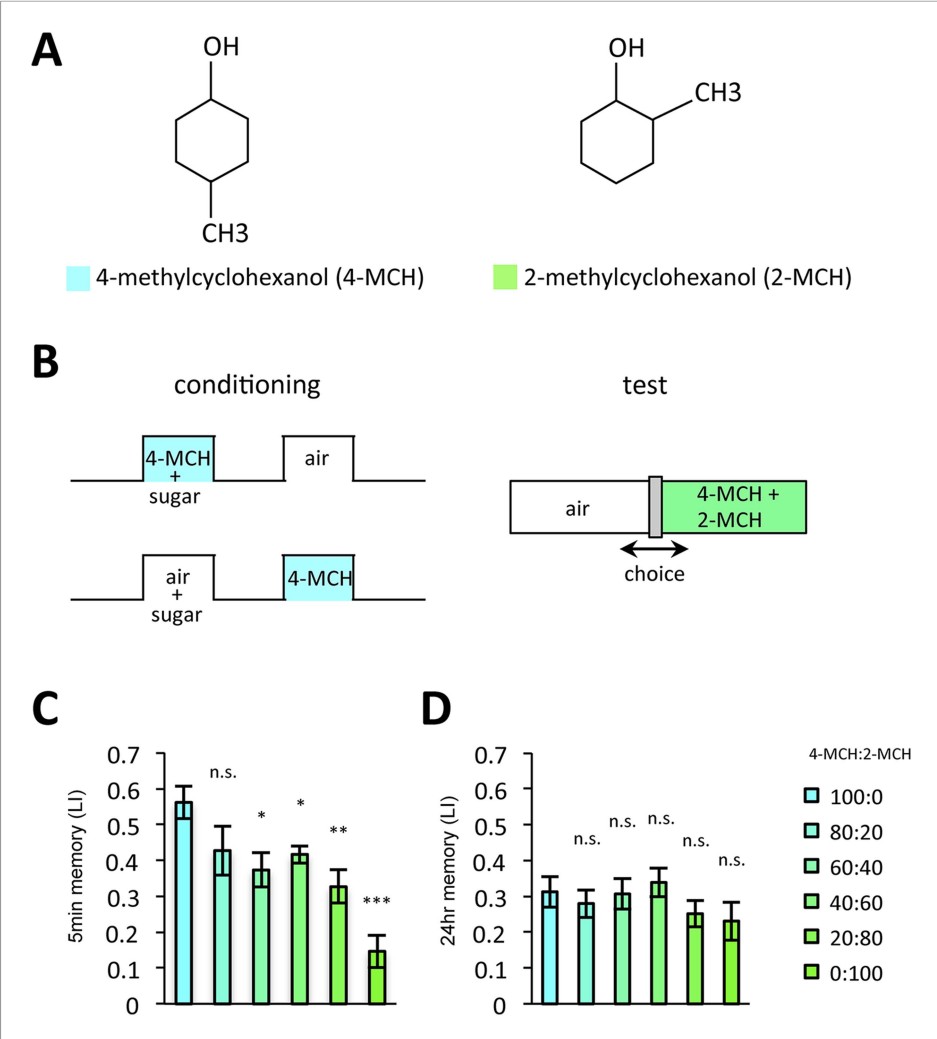

**Figure 10**. LTM specificity is compromised. (**A**) Chemical structures of odorants. (**B**) Design of the experiment. For one group, presentation of 4-methylcyclohexanol was paired with sugar reward. The reciprocal group received sugar without odor. In the test situation, flies of each group were allowed to choose between the air and the mixture of the trained odor and the 'contaminant' (2-methylcyclohexanol). (**C**, **D**) The performance index declined more sharply with the increasing contamination ratio in STM (**C**; $n = 16$ for each) than in LTM (**D**; $n = 24$ for each). Each group was compared to the group that was tested without the contaminant (left most bar). Bar graphs are mean ± s.e.m. *: $p < 0.05$, **: $p < 0.01$, ***: $p < 0.001$, n.s.: $p > 0.05$.

The following figure supplement is available for figure 10:

**Figure supplement 1**. LTM is less specific with another pair of odors.

and inserted into attP40. *c739-GAL4* is an enhancer-trap GAL4 driver described in *Yang et al. (1995)*. Flies were raised at 60% relative humidity at 24℃. Flies for the NMDAR knock down (*Figure 4*) were incubated at 30℃ for 4–5 days prior to the experiment to facilitate the GAL4 activity.

The synaptic blockade experiments (*Figures 3, 7-9*) used F1 progeny of crosses between females of *w;;UAS-shi^ts1* (single copy of *UAS-shi^ts1* from Thomas Preat laboratory, CNRS, France) or *w* and males of the GAL4 drivers. The thermoactivation experiment (*Figure 5*) used F1 progeny of crosses between females of *w; UAS-dTrpA1* (*Hamada et al., 2008*) or *w* and males of *MB310C-GAL4* or *w*. The receptor knock down experiments (*Figures 4, 6*) used F1 progeny of crosses between females of *UAS-Nmdar1^RNAi* (P(TRiP.HMS02200) attP40; Bloomington stock center #41667), *UAS-Nmdar2^RNAi* (P(TRiP.HMS02012) attP2; Bloomington stock center #40846), *UAS-DopR1^RNAi* (P(TRiP.HMC02344) attP2;

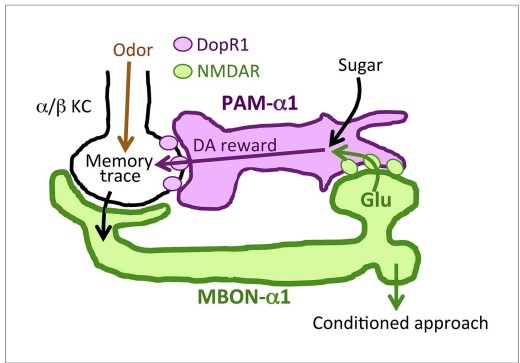

**Figure 11**. Recurrent reward circuit drives appetitive LTM formation. Feedback circuit model of appetitive LTM formation. Neuronal signal of a sugar reward mediated by PAM-α1 converges with olfactory information in the α/β KCs. The coincidental signal is read out through MBON-α1 to give glutamatergic feedback onto PAM-α1 for the gain control of the dopamine release. Appetitive LTM is read out through MBON-α1 to activate other downstream targets.

Bloomington stock center #55239) (*Ni et al., 2011*) or *CS* and males of the GAL4 drivers. The *dumb2* rescue experiments (*Figure 6B*) used F1 progeny of crosses between females of *UAS-DopR1;;dumb2* or *CS* and males of *MB247-GAL4; dumb2* or *dumb2*. *CS* flies were used as wild type (*Figures 6B, 10*).

For immunohistochemistry, the following reporter strains were crossed to the respective GAL4 or LexA drivers: *pJFRC2-10xUAS-mCD8GFP* in *VK00005* (*Figure 1A–D*), *pJFRC200-10XUAS-IVS-myr::smGFP-HA* in *attP18* (*Figure 2A–D*; *Figure 2—figure supplement 1*; *Figure 3B,F*; *Figure 6—figure supplement 1A*) and *pJFRC216-13XLexAop2-IVS-myr::smGFP-V5* in *su(Hw)attP8* (*Figure 2A-E*; *Figure 2—figure supplement 1*), *pJFRC206-5xUAS-IVS-myr-smGFP-FLAG* in *VK00005* (*Figure 1E,F*; *Figure 1—figure supplement 1*; *Figure 3—figure supplement 1A*), *pJFRC51-3xUAS-Syt-smHA* in *su(Hw)attP1* (presynaptic marker; *Figure 1E*), *UAS-CD4::spGFP1-10* and *LexAop-CD4::spGFP11* (*Figure 2F*).

pJFRC2 is described in *Pfeiffer et al. (2010)*. pJFRC200 and pJFRC206 are described in *Viswanathan et al. (2015)* and pJFRC216 in *Nern et al. (2015)*. pJFRC51 was generated by standard methods described in *Pfeiffer et al. (2010)*. *UAS-CD4::spGFP1-10* and *LexAop-CD4::spGFP11* are described in *Gordon and Scott (2009)*.

Multicolor flp-out (MCFO; *Figure 1G*) is a stochastic method that labels individual cells in different colors using a set of three UAS-STOP-epitope constructs that each expresses a different epitope when the STOP cassette is removed. The STOP cassettes in these constructs are each flanked by FRT sites that are removed in a stochastic way by limited expression of flp recombinase. Reagents for MCFO are described in *Nern et al. (2015)*.

## Immunohistochemistry

Dissection and immunohistochemistry of fly brains were done as previously described (*Aso et al., 2014a*). The following antibodies were used: rabbit-anti-GFP (1:1000; A11122; Invitrogen), mouse-anti-GFP (1:200; UC Davis/NIH Neuromab Facility; clone N86/38; for GRASP), mouse anti-nc82 (1:33.3; Developmental Studies Hybridoma Bank, Univ. Iowa, Iowa City, IA, United States) (*Hofbauer et al., 2009*), rat-anti-nCad (1:200; DN-Ex #8; Developmental Studies Hybridoma Bank, University of Iowa), rabbit anti-HA (1:300; Cell Signaling Technology, MA, United States), rat anti-FLAG (1:200; Novus Biologicals, Littleton, CO, United States), mouse anti-Drosophila ChAT (ChAT4B1; 1: 100; Developmental Studies Hybridoma Bank, Univ. Iowa) (*Takagawa and Salvaterra, 1996*), rabbit anti-Drosophila GAD1 (1:1000; a gift from Dr FR Jackson) (*Featherstone et al., 2000*), rabbit anti-DvGluT (1:5000; a gift from Dr A DiAntonio) (*Daniels et al., 2008*) as primary antibodies, and cross-adsorbed secondary antibodies to IgG (H+L): AlexaFluor-488 donkey anti-mouse (1:400; Jackson Labs), Cy3 donkey anti-rabbit (1:500; Jackson Labs), AlexaFluor-647 donkey anti-rat (1:300; Jackson Labs), AlexaFluor-488 goat anti-rabbit (1:800; A11034; Invitrogen, Carlsbad, CA, United States), and AlexaFluor-568 goat anti-mouse (1:400; A11031; Invitrogen).

Confocal images were obtained by using LSM710 (Zeiss, Oberkochen, Germany) (*Figures 1A–F, 2A,E, 3B,F*, *Figure 1—figure supplement 1*, *Figure 2—figure supplement 1*, *Figure 3—figure supplement 1*, *Figure 6—figure supplement 1*) and FV1200 (Olympus, Tokyo, Japan) (*Figure 2F*) and processed by using ImageJ. The magnified images in SIP and SLP (*Figure 2B–D*) were obtained by using LSM880 (Zeiss) with Airyscan and were deconvoluted using the Fiji plugin provided by Bob Dougherty (http://fiji.sc/Deconvolution). Airyscan is equipped with 32 GaAsP detectors that collect light from one illumination spot. It can resolve 140 nm (lateral) or 400 nm (axial) at 488-nm wavelength. For the 3D rendering video, confocal images from LSM710 were deconvoluted using the Fiji plugin and

then were subjected to 3D image rendering software (Volocity: from PerkinElmer, Waltham, MA, United States).

## Behavioral assays

The conditioning and testing protocol was as described previously with minor modifications (*Liu et al., 2012*; *Aso et al., 2014b*; *Yamagata et al., 2015*). Briefly, a group of approximately 50 flies in a training tube alternately received octan-3-ol (3-OCT; Merck) and 4-methylcyclohexanol (4-MCH; Sigma–Aldrich, St. Louis, MO, United States) for 1 min in a constant air stream with or without dried sucrose paper. 3-OCT and 4-MCH were diluted 1% and 2%, respectively, in paraffin oil (Sigma–Aldrich), placed in a cup with a diameter of 14 mm and presented to flies. For thermoactivation experiments (*Figure 5*), 3-OCT and 4-MCH were diluted 10% in paraffin oil, placed in a cup with a diameter of 3 mm and 5, respectively, and presented to flies. For the generalization experiments (*Figure 10*), 1% 1-octen-3-ol (Sigma–Aldrich) and 2% 2-methylcyclohexanol (Sigma–Aldrich) were also utilized. These diluted odors were mixed with the respective ratios. The restrictive temperature for the experiments with *UAS-shi$^{ts1}$* was 32°C and the permissive temperature was 24°C, measured with Venta-Hygrometer (Venta Luftwäscher GmbH, Weingarten, Germany). At the test, the trained flies were allowed to choose between MCH and OCT for 2 min in a custom-made transparent Plexiglas T-maze, illuminated by infrared LEDs (Osram 720-SFH487P, Mouser electronics, Mansfield, TX, United States), and the walls were covered with Fluon (Insect-a-Slip, PTFE30, BioQuip Products, Inc., Rancho Dominguez, CA, United States) and the distribution of flies was imaged by cameras (FFMV-03M2M, Point Grey, Richmond, Canada). Captured images were sent to the computer and the fly number in each tube was counted by a custom-made ImageJ macro. The preference index was calculated by taking the mean indices of the last 60 s in the 2-min choice and the learning index was calculated by taking the mean of preference indices of the two reciprocally trained groups. Half of the trained groups received reinforcement together with the first presented odor and the other half with the second odor to cancel the effect of the order of reinforcement.

For thermoactivation with dTrpA1 (*Figure 5*), flies were trained by being transferred from a background temperature of 24°C to a prewarmed tube in a climate box (30°C) and presented with the training odorant for 1 min.

Flies were aged 3–10 days after eclosion and were starved so that the mortality rate reaches 5–15% at the test. Between the training and the test, flies were kept without food. Conditioning was performed in dim red light and testing was in darkness.

## Statistics

Statistical analyses were performed with StatPlus and Prism5 (GraphPad, La Jolla, CA, United States). Data did not violate the assumption of normal distribution (Shapiro–Wilk test) and homogeneity of variance (Bartlett test). Therefore, the data were analyzed with parametric statistics: one-sample t-test or one-way analysis of variance followed by the planned pairwise multiple comparisons (Bonferroni). The significance level of statistical tests was set to 0.05.

## Acknowledgements

We thank AB Friedrich, A Kucher, S Prech, T Tagawa, and T Onodera for excellent technical assistance; L Nevin, K Tsutsui, K Oyama, B Gerber, and Tanimoto-lab members for critical reading and discussions. We are also grateful to T Preat, J Dubnau, Bloomington stock center, and the TRiP at Harvard Medical School (NIH/NIGMS R01-GM084947) for flies. We also thank members of the Janelia FlyLight Project Team and Janelia Scientific Computing and fly facility for experimental assistance.

## Additional information

### Funding

| Funder | Grant reference | Author |
| --- | --- | --- |
| Deutsche Forschungsgemeinschaft (DFG) | TA 552/5-1 | Hiromu Tanimoto |

| Funder | Grant reference | Author |
|---|---|---|
| MEXT/JSPS KAKENHI | 266646 | Toshiharu Ichinose |
| MEXT/JSPS KAKENHI | 15K14307 | Hiromu Tanimoto |
| MEXT/JSPS KAKENHI | 26840110 | Nobuhiro Yamagata |
| MEXT/JSPS KAKENHI | 25890003 | Hiromu Tanimoto |
| MEXT/JSPS KAKENHI | 26250001 | Hiromu Tanimoto |
| MEXT/JSPS KAKENHI | 26120705 | Hiromu Tanimoto |
| MEXT/JSPS KAKENHI | 26119503 | Hiromu Tanimoto |
| Naito Foundation | | Hiromu Tanimoto |
| Kurata Memorial Hitachi Science and Technology Foundation | | Nobuhiro Yamagata |
| Uehara Memorial Foundation | | Nobuhiro Yamagata |
| Howard Hughes Medical Institute (HHMI) | | Yoshinori Aso, Gerald M Rubin |
| The Strategic Research Program for Brain Sciences | | Hiromu Tanimoto |

The funders had no role in study design, data collection and interpretation, or the decision to submit the work for publication.

## Author contributions

TI, Conception and design, Acquisition of data, Analysis and interpretation of data, Drafting or revising the article; YA, Conception and design, Acquisition of data, Analysis and interpretation of data, Contributed unpublished essential data or reagents; NY, Conception and design, Acquisition of data, Analysis and interpretation of data; AA, Acquisition of data, Analysis and interpretation of data; GMR, Analysis and interpretation of data, Drafting or revising the article, Contributed unpublished essential data or reagents; HT, Conception and design, Analysis and interpretation of data, Drafting or revising the article

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
