## [Decision Letter]

[Editors’ note: a previous version of this study was rejected after peer review, but the authors submitted for reconsideration. The first decision letter after peer review is shown below.]

Thank you for choosing to send your work entitled “Reward signal in a recurrent circuit drives appetitive long-term memory formation” for consideration at *eLife*. Your full submission has been evaluated by K VijayRaghavan (Senior Editor) and three peer reviewers: Leslie Griffith, who is a member of our Board of Reviewing Editors, Martin Giurfa and one other.

The decision was reached after discussions between the reviewers. Based on our discussions and the individual reviews below, we regret to inform you that your work will not be considered further for publication in *eLife* since the amount of work required to make this submission completely solid is too substantial. The complete reviews are appended to help you in revising the work.

In summary, this paper leverages new anatomical reagents to investigate mechanisms of LTM formation for appetitive learning. The finding of an LTM-specific putative positive feedback loop for consolidation involving DANs and MBONs is very interesting. All the reviewers agree that the finding is important, but some of the connections that are made to support the story rest on rather weak or incomplete evidence. Another point of weakness for two of the three reviewers was that many of the key points of support for the new model are based on information that was already available in the literature; there is little new experimental territory explored in this paper. The few novel findings are very suggestive, but not by themselves conclusive or complete enough to warrant publication of this version of the paper. For these reasons the paper in its present form is rejected, but we encourage you to submit a substantially expanded paper when these matters are dealt with (this would of course be treated as a new submission). Specific issues included:

1) Neuronal connectivity: The coexistence in similar spatial domains of terminals from putatively connected neurons is the only evidence of direct connectivity and this is weak. There should be live imaging data (using sensors like GCaMP or EPAC) showing that the neurons in this putative circuit have functional relationships and respond to the postulated transmitters.

2) The evidence that there is no DAergic connection between PAM and MBON needs to be strengthened. To really nail this, the authors should rule out other DARs and be able to show no DA responses in MBON using functional imaging.

3) The nature of the glutamatergic connection (excitatory vs. inhibitory) between MBON and PAM is not investigated. This really matters since this is a synapse that could be inhibitory via mGluR/GluCl or excitatory via AMPA-like or NMDARs. The whole idea that this is an LTP-like excitatory feedback loop, while not inconsistent with data, rests on the assertion that this is an excitatory and potentiating connection. Some evidence is needed here. It is not sufficient for the sign of a synapse to be inferred from indirect behavioral evidence; you need to determine it experimentally. This can be done with functional imaging.

Reviewer #1:

This paper leverages new anatomical reagents to investigate mechanisms of LTM formation for appetitive learning. The finding of an LTM-specific putative positive feedback loop for consolidation involving DANs and MBONs is quite interesting and a bit unexpected. The finding is really novel and important, but some of the connections that are made rest on rather weak or incomplete evidence.

1) Neuronal connectivity:

a) The coexistence in similar spatial domains of terminals from putatively connected neurons is the only evidence of direct connectivity and this is weak. These data are presented as single example images with no quantification or indication of whether these images are typical. At minimum there should be markers showing the presence of dendritic/axonal markers (or citations if these experiments have already been published elsewhere) and GRASP to show contact.

b) There should be live imaging data (using sensors like GCaMP or EPAC) showing that the neurons in this putative circuit have functional relationships and respond to the postulated transmitters.

c) Completeness of model: The authors should be able to show that *shi*^*ts*^ during consolidation in MBON also blocks LTM.

2) The evidence that there is no DAergic connection between PAM and MBON needs to be strengthened – basically a failure of a single RNAi to block LTM. This could be due to: 1) the RNAi not being effective in that cell type, or 2) a different DAR mediating the connection. To really nail this, the authors should rule out other DARs and be able to show no DA responses in MBON using functional imaging.

3) The nature of the glutamatergic connection (excitatory vs. inhibitory) between MBON and PAM is not investigated. This really matters since this is a synapse that could be inhibitory via mGluR/GluCl or excitatory via AMPA-like or NMDARs. The whole idea that this is an LTP-like excitatory feedback loop, while not inconsistent with data, rests on the assertion that this is an excitatory and potentiating connection. Some evidence is needed here. It is not sufficient for the sign of a synapse to be inferred from indirect behavioral evidence; you need to determine it experimentally. This can be done with functional imaging.

Reviewer #2:

This is an excellent work showing for the first time that one-trial appetitive conditioning leading to the formation of long-term memory in *Drosophila* relies on a previously unknown circuit integrated by PAM-α1 neurons, conveying the dopaminergic reward signal to αβ Kenyon cells (KC) of the mushroom bodies (MBs), and MBON-α1 neurons, which are extrinsic to α/β KCs and which provide an excitatory glutamatergic loop onto PAM-α1. In this way, dopamine reward signaling is enhanced via the glutamatergic feedback of MBON-α1, thereby leading to appetitive LTM formation.

In general, I am very enthusiastic about this work and the elegance of the experiments presented. I have a couple of general remarks that the authors could address to improve their manuscript.

I found very interesting that the authors identified two MBON-α1 cells with very similar morphology (Figure 2). Unfortunately, they did not elaborate on the redundancy/complementarity of these two cells for the phenomena under scrutiny. Personally I would like to know why there are two cells. Are they fully redundant? Are they modulating dopaminergic signaling at different rates? Do they contribute equally to LTM formation? Can the authors provide some experimental evidence aimed at disentangling the contributions of the two cells?

Also the circuits presented in Figure 2 and Figure 7, create the wrong impression (in particular 2D, less in the case of 7C) of a closed circuit with no external input/output. What is missing in these figures, and it would be nice to see it, is the octopaminergic circuit proving relay sugar signals to PAM neurons. This is important to see from where does the sugar signal come from. On the other hand, the term “LTM read-out” is obscure. What do the authors mean in terms of circuitry, structures?

The Discussion was a bit poor, in particular given the richness and importance of the results presented in this work. Although it is clear that this would be the first excitatory loop found at the level of the MBs, various reports have studied the presence of inhibitory loops and their involvement in aversive LTM formation in the fly as well as in appetitive memory formation in the honeybee. The latter case is particularly relevant for this article given the common appetitive paradigm. I would appreciate that the authors discuss the parallels and differences between inhibitory, GABAergic feedback neurons, and excitatory, glutamatergic feedback neurons in LTM formation.

Reviewer #3:

The authors report an analysis of a neuronal circuit required for appetitive long-term memory (LTM) formation. A specific type of dopaminergic neurons (PAM-α1) targeting a specific region of the mushroom body is shown to be required for appetitive LTM, but not short-term memory. A connection to a mushroom body output neuron (MBON-α1) is demonstrated, and a feedback loop between MBON-α1 and PAM-α1 proposed. In addition, transmitter release from the specific mushroom body cells targeted by both MBON-α1 and PAM-α1 neurons (α/β-lobe Kenyon cells) are shown to be required during the learning process (acquisition) to induce the formation of LTM.

First, the authors report an anatomical interaction between MBON-α1 and PAM-α1. The cellular anatomy and the determination of the directionality of the signals using a dendrite marker and a presynaptic marker are beautifully described and convincing. However, this finding has been reported already in detail in Aso et al., *eLife* 2014; 3:e04577 (Figure 14N; Figure 20C). The two MBON-α1 cells have been described in this publication to be glutamatergic and not cholinergic or GABAergic as well.

Second, the authors report that blocking the output from PAM-α1 neurons during training does not significantly affect appetitive short-term memory, but LTM. This has been reported already in Yamagata et al., PNAS 2015, 112:578-583 (Figure 4).

Third, the authors show that blocking output from MBON-α1 neurons during training likewise impairs LTM (tested 24h after training), but not STM (tested directly after training). In Aso et al., *eLife* 2014, 3:e04580 (Figure 7) it has been reported that blocking MBON-α1 impairs appetitive 2h memory. Therefore, the finding that LTM, but not STM is affected is novel. However, the conclusion that MBON-α1 neurons mediate specifically LTM (“…we found a similar selective impairment of LTM upon the blockade of MBON-α1”) is perhaps not entirely correct.

Fourth, the authors find that downregulating the dopamine receptor *DopR1* in MBON-α1 neurons does not affect LTM formation or acquisition. However, they find that RNAi-mediated downregulation of *DopR1* in α/β-type Kenyon cells does impair LTM. The authors do not directly exclude that PAM-α1 neurons target MBON-α1, but correctly state that “…these results indicate that the *DopR1*-mediated LTM trace is primarily formed in α/β KCs, not in MBON-α1”. I totally agree with that conclusion, and I find it highly interesting. However, the finding that α/β-type Kenyon cells require this receptor for LTM formation is not surprising given the results from [49], 79: 945-956 and [67], 21: 1647-1653, and excluding a dopaminergic signal from PAM-α1 onto MBON-α1 can be assured only after testing the effect of downregulating the remaining classes dopamine receptors.

Fifth, the authors show that output from α/β-type Kenyon cells is required during the training phase to induce appetitive LTM. This convincing set of experiments confirms the finding obtained by [67], 21: 1647-1653 (Figure 4) that STM and LTM are independently formed in γ-type Kenyon cells and α/β-type Kenyon cells, respectively.

Sixth, it is shown that the mushroom body output neuron MBON-α1 is required for LTM retrieval, whereas the PAM-α1 neuron is required not only during, but also directly after the training. The effect of dopamine released by PAM-α1 is not required 22h after training. This demonstrates that dopamine release does not only signal the CS-US coincidence, but is also required for the induction of LTM shortly after training. This is indeed a novel and interesting finding.

Overall, most of the results presented here have been already described. Only very few aspects are new, which perhaps do not justify the publication of this manuscript in its current form.

[Editors’ note: what now follows is the decision letter after the authors submitted for further consideration.]

Thank you for submitting your work entitled “Reward signal in a recurrent circuit drives appetitive long-term memory formation” for peer review at *eLife*. Your submission has been favorably evaluated by K VijayRaghavan (Senior Editor), Leslie Griffith (Reviewing Editor), and two reviewers.

The reviewers have discussed the reviews with one another and the Reviewing Editor has drafted this decision to help you prepare a minority revised submission.

This study shows that a specific subset of rewarding dopamine neurons in the *Drosophila* brain receives feedback regulation from the developing associative memory trace. This feedback signal is conveyed from the mushroom body by two pairs of mushroom body efferent neurons. This is a novel mechanism of consolidation.

Essential revisions:

We were overall satisfied with the changes made in this paper since the last version that was reviewed. There were two minor remaining concerns that should be addressed in the text in a revised version.

1) There was a bit of concern about the lack of functional imaging, but the data on NMDAR make it highly probable this is an excitatory synapse, but there are no functional data. An acknowledgement that the identification of this connection is preliminary would be appropriate.

2) The wording on OA made it seem like OA is NOT involved. It is, just one synapse back. This should be clarified in the text.

---

## [Author Response]

[Editors’ note: a previous version of this study was rejected after peer review, but the authors submitted for reconsideration. The first decision letter after peer review is shown below.]

We thank the reviewers for constructive comments. In the current manuscript, we addressed all the points by substantially revising the text as well as performing a series of new experiments. The new data added in this manuscript are summarized as follows:

(1) We now performed a GRASP experiment between PAM-α1 and MBON-α1. Strong GRASP signals in the α1 compartment of the MB, SIP and SLP support the contacts between these neurons (Figure 2).

(2) Confocal imaging using a super-resolution detection system revealed that presynaptic bouton-like structures of MBON-α1 are heavily intermingled by PAM-α1 dendrites, again supporting the direct connection (Figure 2).

(3) To gain insight into the nature of the recurrent loop, we now performed a behavioral screening of UAS-RNAi lines for 16 glutamate receptor genes in the dopamine neurons. This screening identified the consistent requirement of NMDA receptor subunits in PAM-α1 (Figure 4). This suggests a positive feedback signal by MBON-α1.

(4) As a further support for the positive feedback, we now provide behavioral evidence showing that transient activation of MBON-α1 can induce appetitive olfactory LTM (Figure 5). To our knowledge, this is the first data showing that the MB output can signal reinforcement, and strongly suggests PAM-α1 as a postsynaptic target of MBON-α1 in the formation of appetitive LTM.

(5) We performed a rescue experiment of a *DopR1* mutant, and found the full rescue of LTM by targeting transgene expression to Kenyon cells (Figure 6). This shows the sufficiency of DopR1 signaling in Kenyon cells, but not the other cell types.

(6) We now demonstrate the requirement of MBON-α1 and α/β KC outputs during early consolidation of LTM (Figure 9). Together with the similar necessity of PAM-α1 in early consolidation, these results strongly suggest the prolonged activity in the recurrent circuit.

(7) We now made a comparison between the generalization profiles of STM and LTM, and found that LTM is less specific to the trained odor (Figure 10). This trade-off between stability and the specificity of memory may well explain the consequence of the prolonged dopamine release without olfactory stimulation in early consolidation.

These new data findings substantially strengthened the previous version of the manuscript, and now offer a more complete picture of the circuit mechanism of appetitive LTM formation. For more than 20 years, physiological studies have postulated that reinforcing neurons in a brain are controlled by learned information both in mammals and in insects (e. g. Schultz et al., 1997, Science; [24], Nature). This study provides an underlying circuit mechanism, which we believe an important contribution to the field.

Please find below our point-by-point response to the reviewers.

Reviewer #1:

This paper leverages new anatomical reagents to investigate mechanisms of LTM formation for appetitive learning. The finding of an LTM-specific putative positive feedback loop for consolidation involving DANs and MBONs is quite interesting and a bit unexpected. The finding is really novel and important, but some of the connections that are made rest on rather weak or incomplete evidence.

1) Neuronal connectivity:

a) The coexistence in similar spatial domains of terminals from putatively connected neurons is the only evidence of direct connectivity and this is weak. These data are presented as single example images with no quantification or indication of whether these images are typical. At minimum there should be markers showing the presence of dendritic/axonal markers (or citations if these experiments have already been published elsewhere) and GRASP to show contact.

b) There should be live imaging data (using sensors like GCaMP or EPAC) showing that the neurons in this putative circuit have functional relationships and respond to the postulated transmitters.

*c) Completeness of model: The authors should be able to show that shi*^*ts*^
*during consolidation in MBON also blocks LTM.*

Our new experiments provided several lines of anatomical and behavioral evidence to clarify this neuronal connectivity. We now better visualize the presynaptic terminals of MBON-α1 enwrapped by the dendritic processes of PAM-α1 by confocal microscopy with the super-resolution detection system as well as the 3D rendered presentation. We also provide several specimens for a stretch of juxtaposition to show stereotypy. To further support this connectivity, we conducted a GRASP experiment by differentially expressing two halves of GFP in the pre- and post-synaptic neurons. All these new data strengthened the anatomical aspect of this feedback circuit.

As MBON-α1 is glutamatergic, we screened the 16 genes encoding different glutamate receptors for the requirement in the dopamine neurons. We found that the knockdown of *dNR1* and *dNR2* in the PAM-α1 caused a selective impairment in LTM. In response to the reviewer’s suggestion, we blocked the output of MBON-α1 and α/β KCs during early consolidation. Similar to the requirement of PAM-α1, these two components of the feedback circuit are necessary for LTM consolidation as well as acquisition, suggesting the sustained activity of this positive feedback loop upon associative training.

We cited [40], Nature, for the neuronal polarity of the PAM cluster neurons, whereas we present the polarity of MBON-α1 in this study (Figure 1). The data are consistent with the mutual feedback between PAM-α1 and MBON-α1. To show the direct monosynaptic connectivity of these neurons, temporally precise presynaptic stimulation with electrophysiological recording of the postsynaptic potential would be necessary. We leave such precise characterization for follow-up studies.

2) The evidence that there is no DAergic connection between PAM and MBON needs to be strengthened – basically a failure of a single RNAi to block LTM. This could be due to: 1) the RNAi not being effective in that cell type, or 2) a different DAR mediating the connection. To really nail this, the authors should rule out other DARs and be able to show no DA responses in MBON using functional imaging.

We agree with the reviewer that the data in the previous experiment does not exclude the possible dopaminergic input to MBON-α1. To back up the argument, we performed a rescue experiment of the *DopR1* mutant LTM by expressing the wild-type transgene in the Kenyon cells. This localized expression fully rescued appetitive LTM, suggesting that *DopR1* signaling in the MBON-α1 neurons is dispensable for appetitive LTM.

As it is not our intention to exclude the contribution of the other dopamine receptors in MBON-α1, we revised the text accordingly and removed the scheme in the previous Figure 4.

3) The nature of the glutamatergic connection (excitatory vs. inhibitory) between MBON and PAM is not investigated. This really matters since this is a synapse that could be inhibitory via mGluR/GluCl or excitatory via AMPA-like or NMDARs. The whole idea that this is an LTP-like excitatory feedback loop, while not inconsistent with data, rests on the assertion that this is an excitatory and potentiating connection. Some evidence is needed here. It is not sufficient for the sign of a synapse to be inferred from indirect behavioral evidence; you need to determine it experimentally. This can be done with functional imaging.

As described above, our RNAi screen for glutamate receptors identified the requirement of *dNR1* and *dNR2* in PAM-α1. Furthermore, depolarization of MBON-α1, just as activation of PAM-α1, was capable to induce appetitive olfactory LTM. These new data are now presented in Figure 4 and Figure 5, and support our model of a positive feedback, while this should formally involve electrophysiological analysis with precise presynaptic stimulation. We also included discussion about this.

Reviewer #2:

*This is an excellent work showing for the first time that one-trial appetitive conditioning leading to the formation of long-term memory in* Drosophila *relies on a previously unknown circuit integrated by PAM-α1 neurons, conveying the dopaminergic reward signal to αβ Kenyon cells (KC) of the mushroom bodies (MBs), and MBON-α1, neurons, which are extrinsic to α/β KCs and which provide an excitatory glutamatergic loop onto PAM-α1. In this way, dopamine reward signaling is enhanced via the glutamatergic feedback of MBON-α1, thereby leading to appetitive LTM formation.*

In general, I am very enthusiastic about this work and the elegance of the experiments presented. I have a couple of general remarks that the authors could address to improve their manuscript.

*I found very interesting that the authors identified two MBON-α1 cells with very similar morphology (*Figure 2*). Unfortunately, they did not elaborate on the redundancy/complementarity of these two cells for the phenomena under scrutiny. Personally I would like to know why there are two cells. Are they fully redundant? Are they modulating dopaminergic signaling at different rates? Do they contribute equally to LTM formation? Can the authors provide some experimental evidence aimed at disentangling the contributions of the two cells?*

We really appreciate for the encouraging comments. As to the two MBON-α1 cells, we did not find any anatomical and genetic evidence that these two cells are different, implying that they are redundant. Our recent systematic characterization of MB output neurons identified that many other MBON types consist of a single pair. This may correspond well to the pleiotropy of MBON-α1 functions. Our work highlighted that MBON-α1, unlike other MBON types, plays a role in all stages of appetitive LTM processing: acquisition, early consolidation and retrieval of LTM. It is therefore reasonable to devise multiple cells of the same function as a fail-safe mechanism. This said, a tool to reproducibly target transgene expression to one of these cells is necessary to show the functional differentiation. Our view above is thus a pure speculation for the moment, and we refrain from mentioning it in the manuscript.

*Also the circuits presented in*
Figure 2
*and*
Figure 7*, create the wrong impression (in particular 2D, less in the case of 7C) of a closed circuit with no external input/output. What is missing in these figures, and it would be nice to see it, is the octopaminergic circuit proving relay sugar signals to PAM neurons. This is important to see from where does the sugar signal come from. On the other hand, the term “LTM read-out” is obscure. What do the authors mean in terms of circuitry, structures?*

We omitted the scheme in previous Figure 2 (see above), as it appears to be confusing. We added the sugar input that leads to the activation of PAM-α1, but we hesitate to add the octopamine input to the PAM-α1 neurons, since octopaminergic reward signals only induce STM, but not LTM ([7], Nature; [29], Curr. Biol.). Following the reviewer’s suggestion, we amended “LTM read-out” to “Conditioned approach” (Figure 11).

The Discussion was a bit poor, in particular given the richness and importance of the results presented in this work. Although it is clear that this would be the first excitatory loop found at the level of the MBs, various reports have studied the presence of inhibitory loops and their involvement in aversive LTM formation in the fly as well as in appetitive memory formation in the honeybee. The latter case is particularly relevant for this article given the common appetitive paradigm. I would appreciate that the authors discuss the parallels and differences between inhibitory, GABAergic feedback neurons, and excitatory, glutamatergic feedback neurons in LTM formation.

We added a discussion on the previous findings of an inhibitory feedback loop in the honeybee MB and contrasted it to our Results.

Reviewer #3:

The authors report an analysis of a neuronal circuit required for appetitive long-term memory (LTM) formation. A specific type of dopaminergic neurons (PAM-α1) targeting a specific region of the mushroom body is shown to be required for appetitive LTM, but not short-term memory. A connection to a mushroom body output neuron (MBON-α1) is demonstrated, and a feedback loop between MBON-α1 and PAM-α1 proposed. In addition, transmitter release from the specific mushroom body cells targeted by both MBON-α1 and PAM-α1 neurons (α/β-lobe Kenyon cells) are shown to be required during the learning process (acquisition) to induce the formation of LTM.

First, the authors report an anatomical interaction between MBON-α1 and PAM-α1. The cellular anatomy and the determination of the directionality of the signals using a dendrite marker and a presynaptic marker are beautifully described and convincing. However, this finding has been reported already in detail in Aso et al., eLife 2014; 3:e04577 (Figure 14N; Figure 20C). The two MBON-α1 cells have been described in this publication to be glutamatergic and not cholinergic or GABAergic as well.

This criticism is not entirely correct. The overlap between different DANs and MBONs reported in our previous paper (Aso et al. 2014 *eLife*) is mainly based on the computational alignment of different samples (image registration), thus the connection is not conclusive. In the current study we provide a direct evidence that these two types of neurons contact in a close proximity using double labelling of PAM-α1 and MBON-α1 in the same brain (Figure 2; Figure 2—figure supplement 1; Video 1). This is further substantiated by our new GRASP data (Figure 2).

We provided the raw data for the polarity of MBON-α1, since they were not presented in the paper. As it is obvious from the comments by Reviewer #1, it is frustrating without information on clear confocal photos and stereotypy. We are however ready to remove these data for polarity and neurotransmitters at the editor’s discretion. This would not compromise the significance of our findings.

*Second, the authors report that blocking the output from PAM-α1 neurons during training does not significantly affect appetitive short-term memory, but LTM. This has been reported already in Yamagata et al., PNAS 2015, 112:578-583 (*Figure 4*).*

This replication experiment of the PAM-α1 blockade serves as a good reference to the same manipulation in MBON-α1 by showing them side-by-side in the same figure. As this is not critical for the story, we are ready to remove these data and just cite the Yamagata et al. paper at the editor’s discretion.

*Third, the authors show that blocking output from MBON-α1 neurons during training likewise impairs LTM (tested 24h after training), but not STM (tested directly after training). In Aso et al., eLife 2014, 3:e04580 (*Figure 7*) it has been reported that blocking MBON-α1 impairs appetitive 2h memory. Therefore, the finding that LTM, but not STM is affected is novel. However, the conclusion that MBON-α1 neurons mediate specifically LTM (“…we found a similar selective impairment of LTM upon the blockade of MBON-α1”) is perhaps not entirely correct.*

We thank the reviewer for the suggestion to make our description more precise. We moderated the wording “selective” to “preferential”.

*Fourth, the authors find that downregulating the dopamine receptor* DopR1 *in MBON-α1 neurons does not affect LTM formation or acquisition. However, they find that RNAi-mediated downregulation of* DopR1 *in α/β-type Kenyon cells does impair LTM. The authors do not directly exclude that PAM-α1 neurons target MBON-α1, but correctly state that “…these results indicate that the DopR1-mediated LTM trace is primarily formed in α/β KCs, not in MBON-α1”. I totally agree with that conclusion, and I find it highly interesting. However, the finding that α/β-type Kenyon cells require this receptor for LTM formation is not surprising given the results from*
[49]*, 79: 945-956 and*
[67]*, 21: 1647-1653, and excluding a dopaminergic signal from PAM-α1 onto MBON-α1 can be assured only after testing the effect of downregulating the remaining classes dopamine receptors.*

As it is not our intention to exclude the contribution of the other dopamine receptors in MBON-α1, we revised the text accordingly.

We agree with the reviewer that the data in the previous experiment does not exclude the possible dopaminergic input to MBON-α1. To back up the argument, we performed a rescue experiment of the *DopR1* mutant LTM by expressing the wild-type transgene in the Kenyon cells. This localized expression fully rescued appetitive LTM, suggesting that *DopR1* signaling in the MBON-α1 neurons is dispensable for appetitive LTM.

The papers the reviewer cited (Perisse et al. and Trannoy et al.) showed the requirement of neurotransmission from α/β KCs for appetitive memory retrieval. Intriguingly, the required Kenyon cells include, but do not necessarily match the cells, where *DopR1* signaling acts or associative plasticity takes place: ([56] Curr. Biol; Zhang et al., 2013 Curr. Biol; Wang et al., 2008 J Neurosci, etc.). Strikingly, Qin et al. showed that *DopR1* expression in γ KCs is fully sufficient to rescue the defect of aversive LTM of the mutant and excluded the role of *DopR1* in α/β KCs for that paradigm, while *DopR1* is strongly expressed throughout the entire MB (Kim et al., 2003, GEP). Thus, our finding on the *DopR1* requirement in α/β KCs is important in cellular distinction of dopamine signaling for appetitive and aversive LTM.

*Fifth, the authors show that output from α/β-type Kenyon cells is required during the training phase to induce appetitive LTM. This convincing set of experiments confirms the finding obtained by*
[67]*, 21: 1647-1653 (*Figure 4*) that STM and LTM are independently formed in γ-type Kenyon cells and α/β-type Kenyon cells, respectively.*

It was unfortunate that our previous writing obscured the point of this experiment. What this experiment shows for the first time is the MB output during memory acquisition is necessary to form LTM, supporting that the loop is functionally active during conditioning. This was not addressed by [67]. We thus included a sentence stating this and expanded the Discussion.

Sixth, it is shown that the mushroom body output neuron MBON-α1 is required for LTM retrieval, whereas the PAM-α1 neuron is required not only during, but also directly after the training. The effect of dopamine released by PAM-α1 is not required 22h after training. This demonstrates that dopamine release does not only signal the CS-US coincidence, but is also required for the induction of LTM shortly after training. This is indeed a novel and interesting finding.

We are grateful for this positive comment. We now included new data showing that output from α/β KCs and MBON-α1 are similarly required during early consolidation (Figure 9), suggesting the sustained requirement of the α1 feedback loop.

Furthermore, we now found that LTM is less specific to the trained odor by comparing the generalization profiles of STM and LTM (Figure 10). This trade-off between stability and the specificity of memory may well explain the consequence of the prolonged dopamine release without olfactory stimulation in early consolidation.

Overall, most of the results presented here have been already described. Only very few aspects are new, which perhaps do not justify the publication of this manuscript in its current form.

We are sorry that our previous version did not articulate the novelty of our results over existing literature. We made substantial revisions of the text for clarification and added new data. Following is the summary of our novel findings:

(1) PAM-α1 and MBON-α1 form a feedback circuit by making mutual contacts both inside and outside the MB (Figure 1 and Figure 2). We provided data showing double labelling of these cell types in the same brain, super-resolution confocal images of them, GRASP signals between these cell types.

(2) Output from MBON-α1 is necessary for LTM acquisition (Figure 3), making the first demonstration of a MBON function beyond memory retrieval.

(3) The NMDA receptor subunits are required in PAM-α1, supporting the glutamatergic feedback from MBON-α1 to PAM-α1 (Figure 4).

(4) MBON-α1 can drive a reward signal for appetitive LTM (Figure 5), suggesting the reward augmentation by the MB-output. Again this function is unexpected for the mushroom body output signal.

(5) The requirement of the dopamine receptor *DopR1* in α/β KCs for appetitive LTM (Figure 6) revealed a clear cellular distinction to *DopR1* signaling for aversive LTM, supporting the idea that the memory trace is formed in the MB-α1 compartment.

(6) The requirement of MB output (α/β KCs) during the acquisition of LTM (Figure 7) has never been shown (and not even a formulated hypothesis to our knowledge). This also suggests that the nascent memory trace in the α/β KCs is ‘read out’ to activate the α1 recurrent circuit.

(7) MBON-α1 is further involved in LTM retrieval (Figure 8). Given the dispensability of PAM-α1 during the test (Figure 8), this is an additional function of MBON-α1 the α1 feedback circuit.

(8) Post-training dopamine release by the α1 recurrent circuit drives LTM consolidation (Figure 9).

(9) For sugar memory, stability is achieved at the cost of stimulus specificity (Figure 10).

[Editors’ note: what now follows is the decision letter after the authors submitted for further consideration.]

Essential revisions:

We were overall satisfied with the changes made in this paper since the last version that was reviewed. There were two minor remaining concerns that should be addressed in the text in a revised version.

1) There was a bit of concern about the lack of functional imaging, but the data on NMDAR make it highly probable this is an excitatory synapse, but there are no functional data. An acknowledgement that the identification of this connection is preliminary would be appropriate.

We agree that physiological demonstration is needed to clarify the sign of the connection between MBON-α1 and PAM-α1. In the revised manuscript, we explicitly acknowledged this. We thought that the positive feedback is the simplest interpretation of our behavioral data (thermo-activation of MBON-α1 [like dopaminergic PAM-α1] induces appetitive LTM; and NMDAR signaling in PAM-α1 is necessary for appetitive LTM), but these data do not exclude the possibility of inhibitory connection. For example, glutamate released by MBON-α1 might be received through inhibitory glutamate receptors, while NMDAR signaling in PAM-α1 might function in the glutamatergic synapses other than with MBON-α1. Thus we left the nature of connection for future studies.

2) The wording on OA made it seem like OA is NOT involved. It is, just one synapse back. This should be clarified in the text.

We do appreciate the work of others revealing octopamine as a reward signal in insect brains, and now modified the text to make it even clearer. Furthermore, we clarified the role of the octopaminergic signal for inducing short-term memory by activating a subset of dopamine neurons (“In Drosophilia, it is considered […] dopamine neurons project”).